# Illness Perceptions and Quality of Life in Childhood Cancer Survivors

**DOI:** 10.3390/cancers17091383

**Published:** 2025-04-22

**Authors:** Adam Kohút, Veronika Koutná, Marek Blatný, Martin Jelínek

**Affiliations:** 1Department of Psychology, Faculty of Arts, Masaryk University, 602 00 Brno, Czech Republic; adamkohut2@gmail.com (A.K.); blatny@phil.muni.cz (M.B.); jelinek@psu.cas.cz (M.J.); 2Institute of Psychology, Czech Academy of Sciences, 602 00 Brno, Czech Republic

**Keywords:** illness perception, quality of life, paediatric cancer, childhood cancer survivors

## Abstract

The perception of illness in general has long been recognised as a significant predictor of the quality of life for childhood cancer survivors. However, illness perception is a relatively broad construct, and little is known, to date, about the association between the specific dimensions of illness perception and the quality of life. The results of this study therefore provide unique insights into which dimensions of illness perception (particularly emotional response, concern, consequences and understanding) may be useful to target in psychosocial interventions to improve the survivors’ quality of life, and for which dimensions interventions may be less effective.

## 1. Introduction

The diagnosis of oncological disease in children and adolescents is a traumatic event, which impacts the lives of not only the children but their families as well. The paediatric oncological disease makes up only 1% of all oncological diseases but is the second biggest factor in the mortality of children 5–14 years old [1]. Although paediatric oncological disease is very serious, the treatment efficacy of paediatric oncological diseases has increased to around 85% in the USA [2] and similar statistics can be found in other countries as well [3]. Despite great advances in treatment, childhood cancer is still a great challenge to patients, their families and medical staff. Because the life expectancy of these patients is increasing, the focus of research shifted from immediate consequences of the oncological disease to long-term consequences, namely cancer survivorship. In this regard, main attention was given to medical factors influencing health-related quality of life (HRQoL), specifically how diagnosis, sex, late effects and other factors influence HRQoL [4,5,6].

In recent years, more and more attention has been given to psychological factors, because medical factors cannot explain all the variance in HRQoL of childhood cancer survivors. Among these psychological factors, illness perceptions are especially important. Illness perceptions originated from the common-sense model [7], which postulates that illness perceptions influence health outcomes, for example, HRQoL. Specifically, illness perceptions, or how a person perceives health threats (various symptoms associated with abnormal functioning), influence his/her behaviour or coping strategies (for example, visiting a doctor or using home remedies), which in turn influence his/her health outcomes (both physical and psychological). This model received support from a seminal meta-analysis [8] and more recently in a meta-analysis concerning cancer patients specifically [9]. The concept of illness perceptions has several dimensions: perceived symptoms, expected timeline, consequences, causes, personal and treatment controllability, perceived concern, understanding of illness, and emotional representation [10]. While some of these dimensions were not present in the original model, they were added through the years, for example, the whole concept of emotional representations of illness.

Studies, to date, have shown that illness perceptions explain a substantial proportion of variance in both adult and adolescent patients with chronic diseases [11,12,13,14] and similar results can be found in adult cancer survivors [15,16,17]. However, the subject of illness perceptions in relation to HRQoL in paediatric oncology patients is still relatively underdeveloped. Several studies focused on illness perceptions in paediatric oncology patients still in treatment. In one of the first articles, five dimensions of illness perceptions were predictive of HRQoL [18]. In this research, consequences, concern, symptoms, illness coherence and timeline were significant predictors of HRQoL, with consequences being the most significant predictor. In a more recent article, symptoms and timeline were significant predictors of HRQoL in paediatric oncology patients, while controlling for sociodemographic and medical factors [19]. This article also analysed illness perceptions in childhood cancer survivors, but only symptoms were significant predictors of HRQoL. Furthermore, the predictive strength of symptoms did not reach statistical significance when parents’ illness perception of consequences was introduced to the regression model. Husson et al. [20] analysed differences in illness perceptions between young adult, middle-aged adult and older adult cancer survivors. They found out that illness perceptions were more negative in survivors who were diagnosed at an earlier age (adolescence), and this group also showed poorer HRQoL.

To the best of our knowledge, there is no study, to date, which analysed illness perceptions in relationship with HRQoL in childhood cancer survivors, while also controlling for demographic and medical factors. In line with the evidence presented above, we hypothesise, that illness perceptions will predict individual dimensions of HRQoL above and beyond demographic and medical factors because there is evidence, that psychological factors are more predictive of HRQoL in cancer survivors than in patients [19]. Furthermore, we hypothesise that positive illness perceptions will predict a more positive HRQoL, mainly in areas such as life satisfaction and psychological functioning.

## 2. Materials and Methods

### 2.1. Sample

The sample presented in this study is a part of the QOLOP project (Quality of Life Longitudinal Study in Paediatric Oncology Patients). The project studies the quality of life in childhood cancer survivors from a longitudinal perspective and started in 2006. The project was approved by the Ethics Committee of the University Hospital Brno (02-300306/EK). Childhood cancer survivors were contacted by the paediatric oncology clinic with an offer to participate in the study. Participants were administered methods in paper-pencil form in the occasion of regular check-ups at the clinic (convenient sampling was performed). All participants and/or their parents were briefed on the aims, procedure and methods of the project and signed informed consent to participate in the study.

The whole sample numbered 163 childhood cancer survivors aged 11–25 (m = 12.99, SD = 8.00). This study was based on the cross-sectional data obtained in the second wave of data collection. The mean time off-treatment was 8.48 years (SD = 2.70). The total number of participants in the second wave of the project was 217, but due to more severe cognitive late effects of some participants or missing data, only 163 were included in the study. The gender was balanced in this sample (54% males) and the most common diagnoses were leukemia and other solid tumours. Only the minority of the sample was treated with CNS tumours (<15%) or experienced more serious late effects (<10%). For those who experienced late effects, in most cases it was a combination of cardiovascular and nephrological organ function disorders, hearing impairments, endocrine issues, emotional difficulties or learning difficulties. For this study, the sample was divided into child’s and adolescent’s group, because the measure used for assessing health-related quality of life (Minneapolis–Manchester Quality of Life Scale) is different for each age group. The sample characteristics for these groups can be found in Table 1.

### 2.2. Methods

The Minneapolis–Manchester Quality of Life Scale (MMQL) was used for the assessment of HRQoL. MMQL is a disease-specific measure of QOL in cancer patients and two age-appropriate versions reflecting different needs and language abilities of different age groups were used in this study. MMQL-YF [21] is designed for survivors from 8 to 12 years old and includes 32 items divided into 4 subscales: outlook on life and family dynamics (e.g., looking forward to the future, *α* = 0.74; ω = 0.75), physical symptoms (e.g., pain, *α* = 0.69; ω = 0.68), physical functioning (e.g., have a lot of energy, *α* = 0.80; ω = 0.79) and psychological functioning (e.g., feeling sad, *α* = 0.67; ω = 0.62). The Cronbach’s alpha for MMQL-YF *α* = 0.61. The MMQL-Adolescent form [22] is intended for survivors older than 13 years and includes 46 items divided into 7 subscales: outlook on life (e.g., happy with life in general, *α* = 0.83; ω = 0.83), physical functioning (e.g., feeling strong and healthy, *α* = 0.76; ω = 0.76), psychological functioning (e.g., worried about health, *α* = 0.83; ω = 0.83), social functioning (e.g., have many close friends, *α* = 0.84; ω = 0.85), cognitive functioning (e.g., difficulty in concentrating, *α* = 0.87; ω = 0.87), body image (e.g., being happy about the way they look, *α* = 0.84; ω = 0.84), and intimate relations (e.g., difficulty in making friend, *α* = 0.62; McDonald’s omega could not be computed because of the low number of items). The Cronbach’s alpha for MMQL-Adolescent form *α* = 0.82 and McDonald’s omega ω = 0.82. Survivors report their HRQoL on a 4- or 5-point scale. The MMQL-Adult form was not available at the time of data collection; therefore, the MMQL-Adolescent version was used without the upper age limit. The MMQL-YF was exceptionally (in five survivors) administered to 13-year-olds according to the recommendation of the clinic staff because a shorter and simpler form was more suitable for them.

Brief Illness Perceptions Questionnaire (BIPQ) was used for the assessment of illness perceptions [23]. It consists of nine subscales: consequences (“How much does your illness affect your life?”), timeline (“How long do you think your illness will continue?”), personal control (“How much control do you feel you have over your illness?”), treatment control (“How much do you think your treatment can help your illness?”), symptoms (“How much do you experience symptoms from your illness?”), concern (“How concerned are you about your illness?”), understanding (“How well do you feel you understand your illness?”) and emotional response (“How much does your illness affect you emotionally? (e.g., does it make you angry, scared, upset or depressed?)”). Each scale is measured by one item on an 11-point scale. The subscale of causes was left out due to the burden to participants. A higher score in each of the dimensions represents a higher degree of specific illness perception. In the timeline subscale, one more answer option was added in the questionnaire set (“My illness is over and does not continue”.) and coded as 11. These responses were afterwards recoded as 0 in the data because they are closest in meaning (a higher score means a more chronic timeline of illness). Due to the nature of the scale, reliability could not be assessed. However, in the original article, test–retest reliability measured after 3 and 6 weeks was acceptable, ranging from *r* = 0.42 (personal control after 6 weeks) to *r* = 0.75 (symptoms after 6 weeks).

All methods were administered to the survivors in the Czech language. The severity of late effects was evaluated according to Common Terminology Criteria for Adverse Events v3.0 so that the most serious of the occurring late effects was decisive to the resulting degree of severity. The evaluation was performed by a physician.

### 2.3. Statistical Analyses

The reliability of methods was assessed by both Cronbach’s alpha and McDonald’s omega because original articles used Cronbach’s alpha, but McDonald’s omega seems to be a better index of reliability for items on the Likert’s scale [24]. The description of relationships between illness perceptions and HRQoL dimensions was measured by Pearson’s correlation. The main hypothesis was tested by hierarchical regression analysis, which in the first step included demographic and medical factors (age, gender, time off- treatment and severity of late effects) as predictors of individual dimensions of HRQoL. In the second step, illness perception dimensions were included as additional predictors and the difference in explained variance between models was observed and tested. All statistical analyses were performed using SPSS 29.

The sample was divided into two age groups based on the version of MMQL. Due to the differences in the number of subscales and items, item wording, as well as the response scale between MMQL_YF and MMQL-Adolescent form, data obtained by this method cannot be merged and we treat them separately.

## 3. Results

The sample of this study consists of 163 participants divided into two age groups. The child’s group consisted of 47 participants and the adolescent’s group consisted of 116 participants. Descriptive characteristics of both groups can be found in Table 1. Descriptive characteristics of illness perceptions for both groups can be found in Table 2. Taken as a whole, childhood cancer survivors included in this study showed positive illness perceptions. They scored relatively low in consequences, timeline, symptoms, concern and emotional response while scoring high in personal control, treatment control and understanding.

The results of the correlation analyses of illness perceptions and individual dimensions of HRQoL can be found in Table 3 and Table 4. All dimensions of illness perceptions were associated with individual dimensions of HRQoL, except for understanding and treatment control in the child’s group (Table 3). Similar results can be found in the adolescent’s group as well; all dimensions of illness perceptions were associated with individual dimensions of HRQoL, except for personal control (Table 4).

The results of hierarchical regression analyses can be found in Table 5 and Table 6 for both groups. In the first step, only demographic and medical factors were included as predictors of individual dimensions of HRQoL and illness perceptions were added to the second models. In the child’s group, the first models explained the statistically significant variance in outlook on life and family dynamics and physical functioning. Demographic and medical factors explained 23.6% of the variance in outlook on life and family dynamics, with late effects (β = −0.29, *p* < 0.05) being the only significant predictor. Demographic and medical factors explained 35.4% of the variance in physical functioning, with late effects (β = −0.45, *p* < 0.01) being the only significant predictor here as well. In the second models for the child’s group, all models were significant, except for psychological functioning. The total explained variance ranged between 44.8 and 63% in significant models. Furthermore, illness perceptions explained significant additional variance in physical symptoms (ΔR^2^ = 0.30, *p* < 0.01) and physical functioning (ΔR^2^ = 0.28, *p* < 0.01). Significant predictors were found only in physical symptoms and included gender (β = 0.42, *p* < 0.05), treatment control (β = 0.37, *p* < 0.05) and concern (β = 0.39, *p* < 0.05). In the first models in the adolescent’s group, the explained variance ranged between 3.2 and 12.6%. Only models predicting physical and cognitive functioning were significant. In both, late effects were the only significant predictor. In the second model for the adolescent’s group, all models were significant, except for social functioning. The total explained variance ranged between 18.1 and 32.3% for the significant models. Furthermore, similar to the first models, illness perceptions explained significant additional variance in all dimensions of HRQoL, except for social functioning. Almost all dimensions of illness perceptions were significant predictors of at least some dimensions of HRQoL, except for timeline, personal control and symptoms.

## 4. Discussion

The goal of this study was to find out if illness perceptions could predict HRQoL of childhood cancer survivors above and beyond demographic and medical factors which are known to be associated with HRQoL, namely gender, age, time off-treatment and severity of late effects. Furthermore, the study aimed to clarify the relationship between illness perceptions and HRQoL, specifically, if positive illness perceptions predict positive HRQoL.

There is some evidence that psychological factors may be more predictive of HRQoL than demographic and medical factors in cancer survivors than cancer patients [19]. This study further supports this claim and is thus in line with evidence from adult cancer survivors [13], adolescents with sickle cell disease [11], youths with inflammatory bowel disease [14], child cancer patients [18,19], and child spinal muscular atrophy patients [12]. This study showed that illness perceptions are predictive of HRQoL while controlling for demographic and medical factors. In this study, illness perceptions predicted individual HRQoL dimensions above and beyond demographic and medical factors. Specifically, in the child’s group, illness perceptions predicted physical functioning and physical symptoms above and beyond medical factors. The additional explained variance ranged from 21.2% for outlook on life and family dynamics to 63% for physical functioning. Contrary to expectations, illness perceptions were more predictive of the physical side of HRQoL dimensions in the child’s group. A possible explanation might be that children focus more on the physical side of their disease because it is subjectively more important and mainly easier to understand than broad and more general areas such as outlook on life or psychological functioning. This is supported by research in the development of children’s concept of illness, which seems consistent with Piaget’s theory of cognitive development [25]. Children in this age conceptualise illness in concrete terms and thus perceive illness in physiological symptoms, rather than broad areas such as psychological functioning or outlook on life. Higher physical symptoms were predicted by female gender, higher personal control (how much control I feel I have over my illness), and higher concern (over my illness). The relationship of gender and concern with lower HRQoL is a stable finding in literature [6,8], but the relationship with higher personal control is contrary to the existing evidence because it should be associated with positive HRQoL [10]. It is possible that children have a less differentiated concept of personal control, specifically, they believe they have high personal control over their illness while experiencing a high degree of physical symptoms. It is further possible that this relationship is not linear and is valid only for part of the sample experiencing a high degree of physical symptoms. This relationship is complicated and warrants further research. However, the results from the child’s group should be taken cautiously, because the sample size did not fulfil Harris’ criteria for regression analysis [26].

In the adolescent’s group, illness perceptions predicted all dimensions of HRQoL above and beyond medical factors, except for social functioning. Additional explained variance ranged from 11.1% for social functioning to 32.3% for physical functioning. The most predictive aspects of illness perceptions were emotional response (how much my illness affects me emotionally) and concern. Higher negative emotional response and higher concern predicted negative HRQoL, which is in line with evidence from cancer patients in general [9]. It further supports the claim that emotional representations of cancer should be included in the care of patients during treatment as well as survivorship. Other significant predictors include consequences (how much my illness affects my life) and understanding (how much I understand my illness), which predicted cognitive functioning. Consequences were found to be an important predictor of HRQoL in other studies with the same population [18,19], but these studies did not analyse individual dimensions of HRQoL. Late effects were also predictive of cognitive functioning, and they are linked to worse neurocognitive functioning [27]. These results suggest a network of relationships among consequences, late effects, understanding and cognitive functioning. Cancer survivors experience late effects, which they perceive as a negative consequence, and this perception leads to understanding their illness as well, and separately, late effects are also linked to neurocognitive functioning. The rather surprising finding from this study is that treatment control (believing treatment will be successful) negatively predicted intimate relationships and social functioning. A possible explanation for this counterintuitive finding is that high treatment control may be linked to more demanding and long-term treatment, which results in long-term hospitalisation and thus also in social isolation, which is represented in lower intimate relationships and social functioning in survivors included in this study.

Recently, an extension of the original Leventhal common-sense model has been proposed in the context of paediatric chronic disease [28]. The common-sense model of parent-child shared regulation (CSM-PC) postulates that although children and parents hold their unique representations of illness, these representations interact and may be reflected in shared illness management or illness consequences. In paediatric oncology, several studies have documented that parents’ representations of illness are often more negative than children’s representations [29,30]. However, studies examining the interaction between children’s and parents’ perceptions of illness are scarce. Given the links between illness representations and HRQoL, future research could explore the interaction of children’s and parents’ perceptions of illness within the theoretical framework of CSM-PC.

The study has some limitations. Firstly, the reliability of BIPQ could not be estimated, because every dimension is represented with only one item and due to the nature of the study, test–retest reliability could not be assessed. Therefore, the reliability was estimated by the results from the original study. Secondly, the results from the child’s group should be taken cautiously because the number of respondents and predictors did not meet Harris’ criterion for regression analysis [26]. The child’s group was analysed separately due to the nature of the MMQL_YF measure. From a clinical perspective, it would be useful to further distinguish between survivors treated in childhood and adolescence in the adolescent group. However, to maintain the robustness of the analysis and to fulfil Harris’ criterion, we did not further subdivide this group.

## 5. Conclusions

Illness perceptions mostly predicted individual dimensions of HRQoL above and beyond demographic and medical factors for both age groups. Several age-specific relationships between illness perceptions and HrQoL were identified. In children, illness perceptions predicted the physical side of HRQoL (physical symptoms and physical functioning) instead of more psychological dimensions (outlook on life and psychological functioning). In adolescents, positive illness perceptions mostly predicted positive HRQoL. The most predictive aspects of positive HRQoL were emotional response and concern. Other significant predictors include consequences and understanding. It can thus be said that illness perceptions significantly contribute to explaining HRQoL of childhood cancer survivors.

## Figures and Tables

**Table 1 cancers-17-01383-t001:** Descriptive characteristics for the child’s and adolescent’s groups.

		Children	Adolescents
Variable		*N* = 47	*N* = 116
Gender *N* (%)	Males	21 (44.7%)	67 (57.8%)
	Females	26 (55.3%)	59 (42.2%)
Current age	m (SD)	12.33 (0.5)	18.33 (2.79)
	Range	11.52–13.7	11.24–25.26
Age at diagnosis	m (SD)	3.71 (2.01)	8.13 (3.93)
Time off-treatment (years)	m (SD)	7.17 (1.55)	9.01 (2.88)
Diagnosis *N* (%)	CNS	5 (10.6%)	19 (16.4%)
	Leukemia	27 (57.4%)	38 (32.8%)
	Other	15 (31.9%)	59 (50.9%)
Late effects *N* (%)	No	30 (63.8%)	49 (42.2%)
	Mild	13 (27.7%)	38 (32.8%)
	Moderate	1 (2.1%)	19 (16.4%)
	Severe	3 (6.4%)	10 (8.6%)

CNS—central nervous system, Other—other solid tumours (33.8% lymphomas).

**Table 2 cancers-17-01383-t002:** Descriptive characteristics of illness perceptions for both groups.

		Children	Adolescents
Variable		*N* = 47	*N* = 116
Consequences	m (SD)	2.45 (2.75)	3.3 (3.07)
Timeline	m (SD)	2.28 (3.21)	3.29 (3.91)
Personal control	m (SD)	7.49 (3.21)	5.91 (3.35)
Treatment control	m (SD)	9.38 (1.38)	9.16 (1.87)
Symptoms	m (SD)	0.81 (1.81)	1.75 (2.62)
Concern	m (SD)	1.04 (1.73)	3 (2.98)
Understanding	m (SD)	5.38 (3.51)	6.38 (2.91)
Emotional response	m (SD)	1.38 (2.1)	2.71 (2.69)

**Table 3 cancers-17-01383-t003:** Correlations between illness perceptions and individual dimensions of HRQoL in the child’s group (8–12 years).

Variable	Outlook on Life and Family Dynamics	Physical Symptoms	Physical Functioning	Psychological Functioning
Consequences	−0.33 *	0.37 *	−0.6 *	−0.26
Timeline	−0.34 *	0.15	−0.45 **	−0.24
Personal control	0.07	0.06	0.34 *	0.04
Treatment control	0.18	−0.06	0.06	0.04
Symptoms	−0.47 **	0.31 *	−0.62 **	−0.45 **
Concern	−0.36 *	0.34 *	−0.53 **	−0.38 **
Understanding	0.24	0.08	0.13	0.04
Emotional response	−0.46 **	0.46 **	−0.54 **	−0.38 **

*N* = 47, ** *p* < 0.01; * *p* < 0.05.

**Table 4 cancers-17-01383-t004:** Correlations between illness perceptions and individual dimensions of HRQoL in the adolescents’ group (13–20 years).

Variable	Physical Functioning	Cognitive Functioning	Psychological Functioning	Body Image	Social Functioning	Life Satisfaction	Intimate Relationships
Consequences	−0.36 **	−0.33 **	−0.3 **	−0.23 *	−0.13	−0.25 **	−0.08
Timeline	−0.34 **	−0.27 **	−0.24 **	−0.28 **	−0.08	−0.26 **	−0.13
Personal control	0.12	0.07	0.08	0.14	−0.08	0.06	0.04
Treatment control	0.06	0.04	−0.04	−0.06	−0.15	−0.03	−0.25 **
Symptoms	−0.37 **	−0.23 *	−0.28 **	−0.08	−0.08	−0.28 **	−0.13
Concern	−0.38 **	−0.31 **	−0.42 **	−0.37 **	−0.13	−0.44 **	−0.13
Understanding	−0.03	0.22 *	−0.04	−0.01	0.1	0.06	0.1
Emotional response	−0.45 **	−0.34 **	−0.43 **	−0.36 **	−0.23*	−0.41 **	−0.2 *

*N* = 116, ** *p* < 0.01; * *p* < 0.05.

**Table 5 cancers-17-01383-t005:** Hierarchical multiple regression analysis for HRQoL dimensions in child’s group (8–12 years).

Predictor	Outlook on Life and Family Dynamics	Physical Symptoms	Physical Functioning	Psychological Functioning
	β	β	β	β
Step 1: Basic and medical factors				
Gender	−0.154	0.333 *	−0.252 *	0.055
Current age	0.165	0.19	0.1	0.038
Time off-treatment	0.24	0.033	0.176	0.104
Late effects	−0.288 *	0.151	−0.449 **	−0.194
	R^2^ = 0.236 *	R^2^ = 0.196	R^2^ = 0.354 **	R^2^ = 0.061
Step 2: Illness perceptions				
Gender	−0.275	0.416 *	−0.273	0.197
Current age	0.205	0.169	0.198	0.091
Time off-treatment	0.24	0.09	−0.095	0.074
Late effects	−0.145	−0.191	−0.257	−0.036
Consequences	0.329	0.08	−0.351	0.178
Timeline	−0.169	0.187	−0.071	−0.214
Personal control	−0.178	0.368 *	0.059	−0.18
Treatment control	0.102	−0.064	−0.065	0.199
Symptoms	−0.232	−0.243	−0.376	−0.337
Concern	0	0.388	−0.154	−0.244
Understanding	0.2195	−0.014	0.021	−0.347
Emotional response	−0.326	0.437	0.231	−0.125
	R^2^ = 0.448 *	R^2^ = 0.496 **	R^2^ = 0.63 **	R^2^ = 0.311
	ΔR^2^ = 0.212	ΔR^2^ = 0.3 **	ΔR^2^ = 0.276 **	ΔR^2^ = 0.251

*N* = 47, ** *p* < 0.01; * *p* < 0.05.

**Table 6 cancers-17-01383-t006:** Hierarchical multiple regression analysis for HRQoL dimensions in adolescent’s group (13–20 years).

Predictor	Physical Functioning	Cognitive Functioning	Psychological Functioning	Body Image	Social Functioning	Life Satisfaction	Intimate Relationships
	β	β	β	β	β	β	β
Step 1: Basic and medical factors							
Gender	−0.173	0.023	−0.17	0.029	−0.052	0.074	−0.167
Current age	−0.051	0.034	−0.069	−0.013	−0.134	−0.103	0.01
Time off-treatment	0.097	0.018	−0.07	0.011	0.03	−0.025	0.043
Late effects	−0.26 **	−0.354 **	−0.1	−0.18	−0.104	−0.107	−0.072
	R^2^ = 0.11 *	R^2^ = 0.126 **	R^2^ = 0.047	R^2^ = 0.035	R^2^ = 0.034	R^2^ = 0.032	R^2^ = 0.034
Step 2: Illness perceptions							
Gender	−0.156	0.039	−0.153	0.029	−0.061	0.082	−0.169
Current age	−0.032	0.08	−0.037	0.029	−0.132	−0.096	−0.022
Time off-treatment	0.101	0.021	−0.059	0.019	0.04	−0.005	0.061
Late effects	−0.095	−0.26 *	0.04	−0.128	−0.106	0.056	−0.024
Consequences	−0.015	−0.268 *	−0.048	−0.074	−0.036	0.034	0.035
Timeline	−0.072	0.134	−0.025	−0.066	0.049	−0.076	−0.065
Personal control	0.035	0.051	0.009	0.083	−0.084	−0.01	0.012
Treatment control	−0.049	−0.097	−0.099	−0.12	−0.225*	−0.09	−0.311 **
Symptoms	−0.146	0.045	−0.11	0.165	0.012	−0.153	−0.071
Concern	−0.141	−0.22 *	−0.217 *	−0.217	−0.013	−0.286 **	0.02
Understanding	0.019	0.263 **	0.058	0.042	0.141	0.136	0.144
Emotional response	−0.268 *	−0.127	−0.264 *	−0.228	−0.241	−0.205	−0.227
	R^2^ = 0.323 **	R^2^ = 0.302 **	R^2^ = 0.287 **	R^2^ = 0.227 **	R^2^ = 0.145	R^2^ = 0.284 **	R^2^ = 0.181 *
	ΔR^2^ = 0.213 **	ΔR^2^ = 0.175 **	ΔR^2^ = 0.24 **	ΔR^2^ = 0.192 **	ΔR^2^ = 0.111	ΔR^2^ = 0.252 **	ΔR^2^ = 0.147 *

*N* = 116, ** *p* < 0.01; * *p* < 0.05.

## Data Availability

The data presented in this study are available on request from the corresponding author due to legal and ethical reasons.

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
