# Peer review of "Illness Perceptions and Quality of Life in Childhood Cancer Survivors"

_cancers, 2025, doi:10.3390/cancers17091383_

Round 1

Reviewer 1 Report

Comments and Suggestions for Authors

This study is part of the QOLOP project (Quality of Life Longitudinal Study in Pediatric Oncology Patients), which started in 2006 and studies the quality of life in childhood cancer survivors from a longitudinal perspective.

This project evaluates the concept of illness perceptions, which have several dimensions: perceived symptoms, expected timeline, consequences, causes, personal and treatment controllability, perceived concern, understanding of illness, and emotional representation. The literature has not evaluated this domain extensively, and it is very interesting from this point of view.

163 childhood cancer survivors aged 11-25 are included and are evaluated with the Brief Illness Perceptions Questionnaire (BIPQ).

The author described the different results of the questionnaire with individual dimensions of HRQoL correlation

There is some evidence that psychological factors may be more predictive of HRQoL than demographic and medical factors in cancer survivors than cancer patients. In this study, illness perceptions predicted individual HRQoL dimensions above and beyond demographic and medical factors. In the adolescent’s group, illness perceptions predicted all dimensions of HRQoL above and beyond medical factors, except for social functioning

The content of this article is very interesting

Comments on the Quality of English Language

In this article, English should be revised. The verb form should be changed sometimes, and commas should be added.

Author Response

Comment 1:

This study is part of the QOLOP project (Quality of Life Longitudinal Study in Pediatric Oncology Patients), which started in 2006 and studies the quality of life in childhood cancer survivors from a longitudinal perspective.

This project evaluates the concept of illness perceptions, which have several dimensions: perceived symptoms, expected timeline, consequences, causes, personal and treatment controllability, perceived concern, understanding of illness, and emotional representation. The literature has not evaluated this domain extensively, and it is very interesting from this point of view.

163 childhood cancer survivors aged 11-25 are included and are evaluated with the Brief Illness Perceptions Questionnaire (BIPQ).

The author described the different results of the questionnaire with individual dimensions of HRQoL correlation

There is some evidence that psychological factors may be more predictive of HRQoL than demographic and medical factors in cancer survivors than cancer patients. In this study, illness perceptions predicted individual HRQoL dimensions above and beyond demographic and medical factors. In the adolescent’s group, illness perceptions predicted all dimensions of HRQoL above and beyond medical factors, except for social functioning

The content of this article is very interesting

Response 1: 

Thank you for taking the time to review our article and for positive feedback about our study.

Comment 2:

In this article, English should be revised. The verb form should be changed sometimes, and commas should be added.

Response 2:

Thank you for taking the time to review our article and for positive feedback about our study.

Reviewer 2 Report

Comments and Suggestions for Authors

The manuscript is interesting and well prepared… but I have some comments:

-please include the consent of the Bioethics Commission+ number

-what were the diagnoses included in the "others" group? It is difficult to compare patients after treatment e.g. for Wilms' tumor or other low-stage cancer and a patient after limb amputation, enucleation, etc.

-in my opinion, the group of young people should be divided into patients treated in childhood and adolescence

-the conclusion should be more concise and not repeat the description of the results and discussions

Author Response

Comment 1:

The manuscript is interesting and well prepared… but I have some comments:

Response 1:

Thank you for your willingness to take the time to revise our manuscript. Your feedback is greatly appreciated.

Comment 2:

-please include the consent of the Bioethics Commission+ number

Response 2:

The study was conducted according to the guidelines of the Declaration of Helsinki and approved by the Ethics Committee of the University Hospital Brno (02-300306/EK) (please see lines 95-96). Informed consent was obtained from all subjects involved in the study.

Comment 3:

-what were the diagnoses included in the "others" group? It is difficult to compare patients after treatment e.g. for Wilms' tumor or other low-stage cancer and a patient after limb amputation, enucleation, etc.

Response 3:

The categorization of the diagnoses into 3 groups (CNS tumors, leukemia and others) was based on the recommendation of the collaborating oncologist. “Others” diagnoses include a wide range of other diagnoses not falling into the group of "CNS tumours" and "leukemia", most commonly lymphomas and reticuloendothelial neoplasms (33.8 % of “Others” diagnoses) but also other oncological diagnoses occurring in childhood. A brief note on the representation of lymphomas in this category has been added to Table 1.

Comment 4:

-in my opinion, the group of young people should be divided into patients treated in childhood and adolescence

Response 4:

Thank you for this important comment. We completely agree that age at the time of cancer treatment can be a significant factor influencing survivors' perceptions of illness. However, to maintain the robustness of the analysis and fulfil Harris’s criterion for regression analysis, we decided to work with the adolescent group as a whole and not to divide it into further subgroups. We accounted for the factor of age at treatment by including age at diagnosis into the regression model as control variable.

Comment 5:

-the conclusion should be more concise and not repeat the description of the results and discussions

Response 5:

We have revised the conclusion to summarise the most important results of our study concisely and clearly.

Reviewer 3 Report

Comments and Suggestions for Authors

The authors undertook an analysis of the issue of illness perception (IP) in the context of cancer, which is widely recognized as a factor in psychosocial adjustment to a cancer diagnosis. The aim of this study was to examine the relationship between individual dimensions of IP and quality of life (QOL) in individuals who survived childhood cancer. The research sample was relatively large and consisted of 163 long-term survivors aged 11–25. In the correlational analysis, all dimensions of IP were associated with specific dimensions of QOL, except for understanding and treatment control. The results of hierarchical regression analysis, controlling for demographic and medical factors, showed that IP predicted individual dimensions of QOL independently of these factors, with emotional response, concern, consequences, and understanding being the most predictive dimensions. The authors conclude that these findings may contribute to more effective targeting of psychosocial interventions aimed at promoting the quality of life of childhood cancer survivors.

The authors are asked to provide the following additional information:

  1. How were late effects monitored (e.g., cardiac, hearing impairments, endocrine issues, emotional difficulties, intellectual problems, etc.)?
  2. Were all patients continuously monitored by an oncology center?
  1. Supplementary information on treatment is needed — especially in the case of brain tumors, where treatment may include surgery only, or additionally radiotherapy/proton therapy or chemotherapy.
  2. Were there any children who underwent HSCT (hematopoietic stem cell transplantation)?
  3. The authors indicate that positive illness perceptions were generally associated with higher quality of life, but only in the adolescent group. The most predictive aspects of positive HRQoL were emotional response and concern, with other significant predictors including perceived consequences and understanding of the illness. According to the authors, personal control in the children’s group and treatment control in the adolescent group were associated with lower quality of life. It is important to note that illness perception is strongly influenced by the home and family environment — the authors should address this point in the discussion. This is particularly relevant when considering the role of treatment control and its association with lower quality of life, as it often reflects the attitudes and behaviors of parents, who significantly influence both children and adolescents.
  4. Additionally, do the authors have information regarding the type of schooling received by the participants — were they attending regular school or receiving home-based education?
  5. I am also interested in the potential impact of parental education level and socioeconomic status on the outcomes obtained in the study — do the authors possess such data?

It would be valuable for the authors to include the survey questionnaires in a supplementary section of the paper.

Author Response

Comment 1:

The authors undertook an analysis of the issue of illness perception (IP) in the context of cancer, which is widely recognized as a factor in psychosocial adjustment to a cancer diagnosis. The aim of this study was to examine the relationship between individual dimensions of IP and quality of life (QOL) in individuals who survived childhood cancer. The research sample was relatively large and consisted of 163 long-term survivors aged 11–25. In the correlational analysis, all dimensions of IP were associated with specific dimensions of QOL, except for understanding and treatment control. The results of hierarchical regression analysis, controlling for demographic and medical factors, showed that IP predicted individual dimensions of QOL independently of these factors, with emotional response, concern, consequences, and understanding being the most predictive dimensions. The authors conclude that these findings may contribute to more effective targeting of psychosocial interventions aimed at promoting the quality of life of childhood cancer survivors.

Response 1:

Thank you for your willingness and time to review our manuscript. Your feedback is greatly appreciated. We have carefully considered all your comments and tried to take them into account in the revision as much as possible.

Comment 2:

The authors are asked to provide the following additional information:

How were late effects monitored (e.g., cardiac, hearing impairments, endocrine issues, emotional difficulties, intellectual problems, etc.)?

Response 2:

The severity of late effects was evaluated by a physician according to CTCAE (Common Terminology Criteria for Adverse Events v3.0) so that the most serious of the occurring late effects was crucial to the resulting degree of severity. In most cases it was a combination of the late effects you mentioned. These details have been added to the sample description (please see lines 109-112).

Comment 3:

Were all patients continuously monitored by an oncology center?

Response 3:

Yes - all survivors were continuously monitored in the paediatric oncology clinic as part of regular post-treatment check-ups.

Comment 4:

Supplementary information on treatment is needed — especially in the case of brain tumors, where treatment may include surgery only, or additionally radiotherapy/proton therapy or chemotherapy.

Response 4:

Unfortunately, we do not have complete details of the treatment received for all survivors included in this study. For less than a third of the sample (29%), these details are missing. Of the remaining 115 survivors, 24 received radiotherapy. As these details are not complete for the whole sample, we do not provide them in the paper.

Comment 5:

Were there any children who underwent HSCT (hematopoietic stem cell transplantation)?

Response 5:

As mentioned above, unfortunately, we do not have complete details of the treatment received for all survivors included in this study. For less than a third of the sample (29%), these details are missing. Of the remaining 115 survivors, 18 received stem cell transplantation. As these details are not complete for the whole sample, we do not provide them in the paper.

Comment 6:

The authors indicate that positive illness perceptions were generally associated with higher quality of life, but only in the adolescent group. The most predictive aspects of positive HRQoL were emotional response and concern, with other significant predictors including perceived consequences and understanding of the illness. According to the authors, personal control in the children’s group and treatment control in the adolescent group were associated with lower quality of life. It is important to note that illness perception is strongly influenced by the home and family environment — the authors should address this point in the discussion. This is particularly relevant when considering the role of treatment control and its association with lower quality of life, as it often reflects the attitudes and behaviors of parents, who significantly influence both children and adolescents.

Response 6:

Thank you for this valuable comment. We completely agree that parents and their perceptions of illness can influence the way their child perceives illness. However, we believe that there is not yet sufficient empirical support for this claim in the literature. We are aware of studies showing that parents often perceive pediatric cancer more negatively than their children.  Of course, this does not necessarily mean that parents do not influence their children's perception of illness. However, we have not been able to find a study that clearly demonstrates that children's perceptions of illness reflect those of their parents. We have therefore attempted to suggest recommendations for future research directions for this topic. Please see lines 296-305.

Comment 7:

Additionally, do the authors have information regarding the type of schooling received by the participants — were they attending regular school or receiving home-based education?

Response 7:

Unfortunately, we don`t know the details about the type of schooling. It can only be assumed that the majority have completed standard schooling, in some cases modified according to specific needs due to medical conditions. Home education was rather exceptional.

Comment 8:

I am also interested in the potential impact of parental education level and socioeconomic status on the outcomes obtained in the study — do the authors possess such data?

Response 8:

While we agree that these factors could also play a role, unfortunately, we do not have data on parental education and SES.

Comment 9:

It would be valuable for the authors to include the survey questionnaires in a supplementary section of the paper.

Response 9:

We understand that the inclusion of a questionnaire battery might be useful to give the reader a better idea of the methods used. However, we have decided not to publish our questionnaire in the supplementary materials because we are not authorised to publish these methods publicly (open access).  Furthermore, we believe that those interested in the BIPQ or MMQL can find the original version by consulting the sources cited in the article.